# Varied Composition and Underlying Mechanisms of Gut Microbiome in Neuroinflammation

**DOI:** 10.3390/microorganisms10040705

**Published:** 2022-03-25

**Authors:** Rai Khalid Farooq, Widyan Alamoudi, Amani Alhibshi, Suriya Rehman, Ashish Ranjan Sharma, Fuad A. Abdulla

**Affiliations:** 1Department of Neuroscience Research, Institute of Research and Medical Consultations, Imam Abdul Rahman Bin Faisal University, P.O. Box 1982, Dammam 31441, Saudi Arabia; waalamoudi@iau.edu.sa (W.A.); ahalhibshi@iau.edu.sa (A.A.); faabdullah@iau.edu.sa (F.A.A.); 2Department of Epidemic Diseases Research, Institute of Research and Medical Consultations (IRMC), Imam Abdulrahman Bin Faisal University, P.O. Box 1982, Dammam 31441, Saudi Arabia; 3Institute for Skeletal Aging & Orthopedic Surgery, Hallym University-Chuncheon Sacred Heart Hospital, Chuncheon-si 24252, Gangwon-do, Korea; arsharma.biotech@gmail.com; 4Department of Physical Therapy, College of Applied Medical Sciences, Imam Abdulrahman Bin Faisal University, P.O. Box 2435, Dammam 31441, Saudi Arabia

**Keywords:** autism, disorders, gut microbiome, neuroinflammation pathogenesis

## Abstract

The human gut microbiome has been implicated in a host of bodily functions and their regulation, including brain development and cognition. Neuroinflammation is a relatively newer piece of the puzzle and is implicated in the pathogenesis of many neurological disorders. The microbiome of the gut may alter the inflammatory signaling inside the brain through the secretion of short-chain fatty acids, controlling the availability of amino acid tryptophan and altering vagal activation. Studies in Korea and elsewhere highlight a strong link between microbiome dynamics and neurocognitive states, including personality. For these reasons, re-establishing microbial flora of the gut looks critical for keeping neuroinflammation from putting the whole system aflame through probiotics and allotransplantation of the fecal microbiome. However, the numerosity of the microbiome remains a challenge. For this purpose, it is suggested that wherever possible, a fecal microbial auto-transplant may prove more effective. This review summarizes the current knowledge about the role of the microbiome in neuroinflammation and the various mechanism involved in this process. As an example, we have also discussed the autism spectrum disorder and the implication of neuroinflammation and microbiome in its pathogenesis.

## 1. Introduction

Inflammation is part of a defense response that strives to protect us; however, the alterations in the inflammatory response are deemed responsible for causing various diseases. The inflammatory cascade has come under the radar due to the mounting impact of inflammation on our daily health. The changing lifestyle, diet, and other environmental factors have added to the inflammatory burden in our daily lives. The resulting research has implicated a dysregulated inflammatory response in neurodegenerative and neuropsychiatric diseases such as depression [1]. This inflammation has been termed neuroinflammation.

Neuroinflammation is the inflammatory response occurring in the nervous system and involves various immune mediators [2]. The key players of this process are cells and cytokines, mainly microglia. In addition, other immune cells such as endothelial cells and astrocytes also play important roles [3]. Neuroinflammation is a critical physiological process that communicates between the central nervous system (CNS) and the immune system. It plays an essential role in brain development and plasticity, maintenance of synapses, repair processes after spinal cord injury or traumatic brain injury [4]. Neuroinflammation is an integral part of the pathology of neurodegenerative disorders, including Parkinson’s disease (PD), Alzheimer’s disease (AD), functional disorders, major depressive disorder (MDD), bipolar disorder, schizophrenia as well as autoimmune disorders, such as multiple sclerosis (MS) [4]. A host of physical factors governs the occurrence and regulation of neuroinflammation, and the human microbiome is one of the most important ones, as discussed under.

The human microbiome has been implicated in neuroinflammation in recent times. Changes in the microbiome, especially that of the human gut, can have subtle effects on the individual’s brain cognitive capacity and behavior [5]. The changes in the activity of various brain regions in response to microbiome changes suggest that human microbiota and associated products are important determinants of neuronal coordination [6]. Due to the impact of the microbiome on brain function and activity, its role in determining the behavioral responses and depression has been dramatically speculated and studied (Figure 1). Studies conducted on the animal models with altered commensal intestinal microbiota or pathogenic bacteria show that their behavioral responses are altered in response to changes in the gut microbiota [7]. Owing to its neuromodulatory effects and its role in depression, anxiety, and stress response, the microbiome can serve to identify new drug candidates and techniques for treatment-resistant depression (TRD) [8,9]. Various probiotics have shown promising results in animal model studies for treating anxiety and depression [5]. Some probiotics have also been tested for their role in reducing depression and anxiety in human subjects [8,10]. However, the data available to date are limited, and in-depth research is needed to further our understanding of the role of these probiotics in treating depression and anxiety.

Human microbiota or gut microbiota is the plethora of microorganisms with a higher amplitude of bacteria inhabiting primarily on the surface of the oral mucosa or of the intestines. The microbiome and the host organism maintain a mutually beneficial or symbiotic relationship [11]. The genome of these microscopic creatures is estimated to be more than the genome of the host under the radar, the human [12]. Recent studies indicated that the human body accommodates approximately 3.8 × 10^12^ bacterial cells, with a wide range of them colonizing in the human intestinal tract [13]. However, microbiota comprises bacteria, and other forms of microorganisms such as fungi, archaea, protozoa, and viruses are also accommodated. The microbiome possesses the capability of regulating the development of the brain, dynamics of immunity and blood pressure, and metabolic activities in the body through the microbiome–gut–brain axis [14]. In the human microbiome, the bacteria surpass fungal and archaeal microorganisms by 2–3 folds of magnitude and dominate others [15]. In recent years, there has been a significant awareness of the microbiome for its attractive benefits to human health [16]. Sender et al. have reported the occurrence of 3.8 × 10^13^ bacterial cells in the gut microbiome (GMB) and 3.0 × 10^13^ human cells in a human subject of 70 kg [13]. An enormous variety of bacteria has been seen among healthy individuals. However, some bacterial phyla dominate the vast microbial community, for example, *Bacteroidetes*, *Actinobacteria*, *Proteobacteria*, and *Firmicutes* [17]. The community of these microbes differs significantly among humans, primarily depending on varying body habitats, starting from the neonatal stage unto the whole lifespan, based on lifestyle and the disease condition. Some of the most familiar existing bacterial species include *Cladosporium* spp, *Helicobacter pylori*, *Lactobacillus bulgaricus*, *Lactococcus lactis* subspp. *Cremoris*, *Bacteroide intermedius*, *Bifidobacterium*, *Streptococcus mutans*, *Clostridium* spp. [18]. The microbiome’s diversity and precise composition differ depending on factors, including human genetics and immune interactions during early development, lifestyle, food, and antibiotics’ use [19]. The changes in the microbiome have been correlated with various pathologies, including major depressive disorder [20], inflammatory bowel disease [21], and cancer [22]. The microbiome can trigger the immune system into alternative states and has been linked to the pathology of various autoimmune diseases [23].

## 2. Signaling Mechanisms at Brain–Gut–Microbiota (BGM) Axis

Trillions of commensal microorganisms (Microbiota) house different areas of the host, which are exposed to an external environment. Around 1013 to 1014 species of microbes reside in the gut; together with their metabolites and genomic materials, they form the microbiome [24] and spread alongside the intestinal villous epithelium layer [25]. Among the population, the composition of the microbiota is interindividual varied [17]. In healthy adults, the composition is relatively stable and diverse [24,26]. However, the composition tends to be dynamic and less diverse during early-life infancy to toddler age and in elderlies [26,27].

The gut microbiome is in constant bi-directional communication with the central nervous system (CNS) through different networks forming thegut–microbiota-brain (GMB) axis. Numerous systems are acting collaboratively to ensure the efficiency of this axis, including CNS, enteric nervous system (ENS) arm of the autonomic nervous system (ANS), and endocrine and immune systems [7,28,29]. The gut transmits signals to the brain by three main routes, including neuronal channel (represented by afferent vagal fiber/spinal cord), neuroimmune (via cytokines), and neuroendocrine (via hypothalamus pituitary adrenal (HPA) axis) pathways [30,31,32]. These communications are mediated by the gut microbiota’s metabolites, which aim to influence CNS development, function, and physiology [33]. Comparably, the CNS utilizes hormonal, immune, and neuronal pathways to stimulate gut-related functions, including motility, secretory, intestinal barrier permeability, and gut mucosa immunity [34].

Most of these communication-mediated metabolites, peptides, hormones, and endotoxins are sensed locally by the afferent vagus nerve (VN) receptors, representing the direct path to convey signaling inputs to the CNS [34]. Specifically, VN receptors get activated by signaling molecules (such as serotonin, gut hormones, and cytokines) and transmit generated singles to the brainstem [14,35]. The other chunk of these bioactive molecules may cross the intestinal barrier to the blood circulation, where they might reach the brain and cross the blood–brain barrier (BBB) [29].

The VN comprises 80% of the afferent fibers and 20% of the efferent fibers [24]. It acts as a superhighway for information passaging in the brain-gut crosstalk [29]. The afferent fibers innervate the gut mucosa yet do not directly communicate with the luminal microbes [24]. Therefore, these fibers can receive indirect and direct signals using different mechanisms. For instance, microbiota metabolites such as long-chain fatty acid (oleate) trigger the VN receptor through CCK-mediated means, while short-chain fatty acids (SCFAs), such as butyrate, directly influence the VN terminal [36]. Moreover, VN receptors interact with the secretion (hormones and/or neuroactive molecules) from metabolites-activated gut-sensory cells such as the enteroendocrine cells (EECs) [29]. Thus, the VN is considered a critical neuro-anatomical path that connects GIT to CNS [37].

EECs are one of the resident mucosal cells (<1%) that consist of thousands of dispersed cells [38] located in the gastrointestinal tract (GIT) and distributed along the gut mucosa, and yet, they represent a large number of endocrine cells [39,40]. They can sense luminal contents, whether its metabolites or endotoxins [40]. Furthermore, they contained more than twenty hormones that modulate food consumption, GI motility, and secretion via the enteric nervous system (ENS) [35,41,42]. The microbial metabolites stimulate the release of the EECs-hormones, which initiate a signal that acts on the afferent vagal fiber receptors further convey these stimuli to the brainstem, the intermediate station of the bi-directional axis [41,43]. For instance, spore-forming *clostridial* gut microbiota metabolizes the indigestible fiber-rich carbohydrate and primary bile acid to produce SCFAs and secondary bile acid (2BAs), respectively [30,44]. SCFAs mediate the microbiota communication by acting on the receptors of enterochromaffin cells (ECCs), a type of EECs [29]. Additionally, In mixed colonic cultures in vitro model, SCFAs activate G-protein–coupled free fatty acid receptor (FFAR2) of the intestinal enteroendocrine L cells and regulate the secretion of glucagon-like peptide (GLP-1) and peptide YY (PYY) hormones [45,46]. The latter has a role in food intake reduction, while the former influences blood sugar homeostasis via carbohydrate and fat ingestion [37,47]. Moreover, 2BAs were reported to promote the secretion of GLP-1 enteroendocrine hormone [48]. Suggesting that, Intestinal hormones can influence the CNS throughout the vagal–brainstem–hypothalamic route [37]. Moreover, SCFAs such as butyrate, acetate, deoxycholate, and propionate increase the biosynthesis of 5-hydroxytryptamine (5-HT), known as serotonin, in the colon ECCs [46,49].

Serotonin is one of the tryptophan metabolites and considers a key neurotransmitter in both terminals of brain–gut communication [50]. A reduction in serotonin level was observed in serum and gut of germ-free (GF) mice. This reduction was restored when the mice were populated with specific pathogen-free (SPF) microbiota [46]. The microbiota can control the tryptophan metabolism through the kynurenine pathway (KP), which is influenced by inflammatory mediated enzymes such as indoleamine-2,3-dioxygenase (IDO1) [31,51]. Hence, Microbiota can reduce the tryptophan fraction for serotonin production and increase the synthesis of other KP metabolites [31,50]. Two sources produce the enteric serotonin, the mucosal ECCs predominantly and serotonergic neurons with less quantity [52]. Mucosal serotonin permits a pro-inflammatory reaction in the ENS, while neuronal serotonin delivers neuroprotection [52]. Synergistically, mucosal and neuronal serotonin maintain the GI function [52]. Accordingly, the gut microbiota influences both serotonergic and tryptophan metabolism [50].

In the same way, the microbiome displays intimate communications with mucosal immune cells. The gut microbiome’s impact on the immune–neural interaction occurs in an early stage of life [25]. Educating the defense system of the host organism provides a balanced function between tolerance and immunity, which delivers host homeostasis [35,53]. The immune cells inhabit the lamina propria and are organized in different structures such as Peyer’s patches and mesenteric lymph nodes (MLN) that are collectively called the gut-associated lymphoid tissue (GALT) [54]. In total, 70% of the immune system represented by the GALT resides in the gut. Moreover, the GALT housed about 80% of the immunoglobulin A (IgA) plasma cells [53]. The IgA influences the immune system’s homeostasis and other physiological processes by maintaining its interaction with the gut microbiome [55].

In addition, microbiome contact with innate and adaptive immune cells across the intestinal mucosa further impact the homeostasis of immune-related entities, such as GALT and CNS-immune cells [25,56]. Manipulation of gut microbiota modifies the neuroimmune system, specifically during microglia development and maturation [25]. In a germ-free (GF) mice study, microglia from newborn and adult stages were isolated for gene expression profile analysis. Compared with matched-age control mice, the adult microglia showed high numbers of downregulated genes (around 322 genes) compared with the newborn. These genes were associated with inflammation, defense mechanisms, and the late adult signature of microglia development. These results exhibit microbiome involvement in microglial pre-phenotype transition, adult-phenotype, and immune signaling [57]. Furthermore, microglial immaturity and malformation in GF mice can be restored by SCFAs [58]. Additionally, the SCFAs signaling molecules, butyrate, showed induction of regulatory T cells (Treg) differentiation in the lamina propria of SPF mice colon and further increased interleukin-10 (IL-10), an anti-inflammatory cytokine, production by the Treg. This Treg induction was exhibited in both models in vitro and in vivo [59]. On the other hand, The Fox3+ Treg cells controlled the microbial balance by suppressing the inflammation and regulating IgA-secretion in Peyer’s patches [60].

Studies have shown that toll-like receptors (TLRs) regulate intestinal homeostasis by working as a recognition receptor for microbial ligand and expressed in the ENS [61,62]. TRL4 recognition of endotoxin, such as lipopolysaccharide (LPS), stimulates pro-inflammatory cytokines production via the NF-κB pathway [61]. Moreover, TLR2 can control the integrity of the ENS by regulating intestinal inflammation [63]. ENS (second brain) interfaces with EECs and GLAT and may communicate with the gut microbiome via afferent nerves such as the VN [43]. Remarkably, the immune system, microbiome, and nervous system are intricately interactive and communicate through secondary metabolites, cytokines, neurotransmitters, and their corresponding receptors [25].

As previously mentioned, the gut-microbiota can regulate the host’s immune activity by regulating the pro-inflammatory cytokine production, which subsequently influenced the hypothalamic–pituitary–adrenal (HPA) axis to release their corresponding hormones CRH, ACTH, and cortisol, respectively [31]. The neuroendocrine stress system represented by the HPA axis corresponds to the other side of the BGM axis (from the brain to the gut). Moreover, the VN provides an essential line of communication between the HPA and the gut microbiota [64]. The HPA axis is modulated by physical stressors such as infection and psychological stress [65]. As mentioned earlier, the TLRs recognize the pathogenic element leads to a cascade of events, including activation of NF-κB pathway, cytokines production, and eventually HPA response [64]. Moreover, the stress-mediated HPA response can be regulated by neurotransmitters, such as serotonin [66].

Stress, gut microbiome, and inflammation factor can alter the integrity of the two critical barriers in theGMB axis, the intestinal barrier, as well as the BBB [29,54]. In normal conditions, the gut epithelial layer provides a physical barrier to prevent any unregulated translocation of the gut microbiome to the gut lamina propria [67]. However, some environmental factors may influence the intestinal barrier and affect its integrity [54]. For instance, psychological stress provokes gut hyperpermeability, allowing the translocation of bacteria and some microbial antigens to the gut mucosa, which triggers the differentiation of T cells and B cells (activate the mucosal immune response and pro-inflammatory cytokines release) and derives the HPA axis activation [64,67]. In an in vitro study, the induction of pro-inflammatory cytokine, interferon γ to T84 and Caco-2 monolayers (human colonic monolayers), affected the permeability of the gut epithelium. Further, it allowed the translocation of commensal E.coli via lipid raft-mediated transcytotic paths [68]. Alternatively, the effect of SCFAs mixture on the intestinal barrier’s function was evaluated using the Caco-2 Bbe1 cells model [69]. A moderate concentration of SCFAs mixture with high butyrate percentage promotes the protection and repair of disturbed intestinal monolayer barrier upon exposure to inflammatory factors such as LPS/tumor necrosis factor-α [TNF-α] [69]. Moreover, probiotic treatment (*Lactobacillus farciminis*) in acutely stressed rats exhibited stress-prompt colonic hyperpermeability (leaky gut) attenuation. Moreover, it inhibited HPA-axis stress activation (neuroendocrine response) and pro-inflammatory cytokines expression [70].

The BBB is considered as another level of microbiota-CNS communication. Similarly, the CNS is protected from circulating proteins via vascular endothelial cells along with glial cells forming the BBB, which provides an astringent, highly selective barrier that permits controlled nutrition and communication [67]. In a leaky gut, microbes can translocate to the gut mucosa or release factors in the blood circulation [67]. These circulating molecules, including endotoxin and indigestible metabolites, may activate the peripheral immune cells to trigger chronic and systemic inflammation and/or interact with the BBB [67,71]. Although studies of systemic inflammation induce BBB disturbance was modulated via LPS, the evaluation of the LPS in vivo studies showed only 60% of poor BBB integrity [72]. Claudins and occluding are the proteins that, in physiological conditions, maintain tight junctions supporting the barrier [67] and show alterations in expression in case of infection [30]. However, a current study showed that oral admission of amoxicillin-clavulanic acid (antibiotic) altered the gut microbiome composition and increase the BBB permeability in rhesus monkeys [73]. Moreover, microbial metabolites can influence the BBB integrity by triggering the brain glial cells, for example, astrocytes and microglia [58,73].

Gut inflammation and BGM axis disturbance (gut dysbiosis) are linked to many metabolic and neurological disorders [74]. Therefore, targeting critical hubs of the GMB network such as SCFAs, serotonergic pathways, VN, and CNS macrophages holds tremendous therapeutic and investigation benefits to ameliorate the consequence of gut dysbiosis.

## 3. Neuroinflammation

A broad range of chemokines, cytokines, secondary messengers, and reactive oxygen species (ROSs) contribute toward the overall inflammatory response inside the nervous system and related tissues. CNS resident glia (including both microglia and astrocytes), endothelial cells, and immune cells derived from the peripheral sources produce the aforementioned inflammatory mediators [75]. Although neuroinflammation is usually associated with disease pathogenesis of various neurological disorders, it is primarily a protective phenomenon and has protective, beneficial, and adaptive effects [76]. Firstly, neuroinflammation is the communicator between CNS and the immune system [77]. This can, in turn, help in a coordinated response that propagates through the production of cytokines and secondary signals, leading to appropriate behavioral and physiological changes, including fever, lethargy, hypophagia, decreased activity, and reduced social interaction [78]. These changes, in turn, help in reallocating the host resources to fight the infection [79]. Through a complex interaction, the cytokines orchestrate brain development and plasticity [80,81,82] by learning and memory during long-term potentiation (LTP) [83,84]. IL-1β, IL-6, and TNFα, the significant components of neuroinflammation, have been implicated in maintaining synapses [85,86,87]. In addition, IL-1β bolsters the learning process, whereas IL-6 inhibits it [88]. Neuroinflammation is also an essential part of the repair processes after spinal cord injury or traumatic brain injury by maintaining a balance between nascent and active states of the microglia [89]. Together, the involvement of neuroinflammation in these processes indicates that it is a part of the normal physiological processes occurring in the body. Under physiological circumstances, this response is short-lived and is quickly resolved after restoring homeostasis. However, it is important to explore its mechanisms in detail so that the prolonged inflammation, the eventual matter of concern, may be understood.

Microglia are regarded as one of the essential players in the act of neuroinflammation [90,91,92,93], also called “policemen of the brain” [94]. They also propagate the inflammatory signals that originate in the peripheral parts of the body and tend to coordinate the immune system and the brain. The main goal of these CNS resident macrophages is the protection of CNS; however, under certain circumstances, the chronic or exaggerated amplification of these cells and their effector functions can lead to various pathological conditions such as neuronal injury and death as well as neurocognitive disorders [95]. Chronic or traumatic stressors promote a pronounced neuroinflammatory profile [96]. Animal model studies conducted on rodents suggest that in response to chronic or traumatic stressors, such as repeated social defeat and foot shock, various inflammatory signals are released by the macrophages, particularly in the stress response regions of the brain [97,98,99]. The infamous chronically damaging neuroinflammation and the vulnerable newborn neuronal population sustains damage even from even the minute imbalances and prolonged inflammatory events, even before the detectable rise in peripheral cytokines concentration, can induce a behavioral change [100]. These newborn neurons are essential for maintaining mood, normal functioning of the brain, and recovery from depression through the effect of antidepressant medications [101,102]. The oxidative stress due to inflammation and injury and production of reactive oxygen species further compounds inflammatory processes and often leads to neurodegeneration [103,104,105,106,107,108]. However, various studies indicate that this is a two-way road, whereby the anomalies in the immune system together with genetic and environmental cues can trigger neuroinflammation, and this neuroinflammation can then stimulate the activity of inflammatory pathways [109,110]. A positive feedback loop thus ensues, and a vigorous cycle thus ensues, leading to ongoing neuronal loss and decline in the overall health of CNS.

Microglia age, leading to disrupted brain immune communication and imbalance of several mediators of immunity [111]. The morphological profiles of the aged microglia are also distinct with characteristic de-ramification, fewer branching arbors, shorter processes that leave them indifferent to regulatory stimuli, and the persistence of the pro-inflammatory profile of microglia long after an injury can have a detrimental impact [112,113,114]. The increased expression of systemic inflammatory mediators with aging can also have profound effects on neuroinflammation. These changes can lead to the chronic activation of the perivascular and parenchymal microglia with distinct pro-inflammatory cytokines’ expression profiles [115,116], in turn, contribute to an overall increase in the vulnerability to neuropsychiatric disorders. In addition to aging, the lifestyle and overall health status are also significant predictors of the neuroinflammatory status directly associated with disease risk and pathogenesis of brain and nervous system-related disorders [117]. The physical health status is one such regulator of neuroinflammation. In a study conducted on a cohort of obese women, increased levels of various pro-inflammatory markers, including IL-6, C-reactive protein (CRP), and adipokines, were correlated directly with the symptoms of anxiety and depression [118]. Another study reported that anxiety was alleviated in their subjects, corresponding with reducing inflammation after fat removal surgery [119]. Aging-related diseases of the cardiovascular, metabolic, neuroendocrine, cerebrovascular, and immune systems increase the risk of major depression [120]. The inflammation has also been linked to the resistance to conventional antidepressant treatments [121,122,123]. The putative mechanisms that link inflammation with depression include oxidative stress, imbalance of inflammatory cytokines [124], and hyperglutamatergia [125,126,127]. Unlike full-blown inflammatory disorders, the neuroinflammation involved in cognitive processes is a slow process that thrives on vulnerabilities and environmental stimuli [128]. The inflammatory challenges can act as triggers to the pre-existing genetic risk for various neurodegenerative disorders. The inflammatory triggers and genetics of the susceptible individuals can lead to disease onset and pathogenesis [129].

In some cases, certain viruses or bacteria alter the standard immune system responses to environmental challenges [130]. In addition to the microglial activation and production of molecular immunological mediators, certain other molecular, cellular, and physiological events have been associated with neuroinflammation. Among these, movement of immune cells from the periphery into the CNS, increased blood–brain barrier (BBB) permeability, and brain edema are the most prominent features associated with neuroinflammation progression [131]. However, these changes are usually observed at many later stages. Together, these processes lead to ischemia, vascular occlusion, and cell death, contributing to the loss of neurons, tissue microenvironment, and homeostasis within the CNS.

Neuroinflammation, thus, appears to be a necessary physiological process. However, prolonged, unchecked, and unregulated neuroinflammation can trigger pathways that may lead to nervous system-related disease manifestations such as anxiety, depression, memory loss, and cognitive impairment. Thus, understanding the neuroinflammatory pathways can help solve the puzzle of pathogenesis as well as aid in designing new drug candidates with more specificity, lesser side effects, and better disease prognosis of these disorders.

## 4. Microbiome Modulates Inflammation

Whipps and colleagues coined the term microbiome in 1988 to be a “characteristic microbial community” in a “reasonably well-defined habitat which has distinct physicochemical properties” as their “theatre of activity” [132]. With the advancement in microbiology and research, several modifications have been made to this definition. The current, most cited definition describes microbiomes as a community of commensal, symbiotic, and pathogenic microorganisms within a body space or other environment. Berg and colleagues have recently redefined the microbiome as a “characteristic microbial community occupying a reasonable, well-defined habitat with distinct physicochemical properties” [133]. According to this definition, the microbiome is not limited to microorganisms, but it also encompasses their activities, which together constitute specific ecological niches. These micro-ecosystems are usually integrated into macro-ecosystems, including eukaryotic hosts, where they perform various necessary functions for the host’s functioning and health.

The human microbiome is vital and is now even considered the “last organ” [134]. There are great variations in its composition from person to person and site to site, namely in nasal passages, skin, oral cavity, urinogenital tract, and gastrointestinal tract [18]. It is not limited to bacterial species but also contains viruses, skin mites, eukaryotes, and archaea. An estimated 500 to 1000 bacterial species exist in our body, primarily in the mucosal and skin environment [135,136]. The composition of the human microbiome is determined by genetic and environmental factors during the early development, diet, lifestyles, and exposure to antibiotics. These factors determine the eventual consequences of such interactions and emergence of disease, including inflammatory bowel disease [129,137,138,139,140,141], cancer [142,143,144,145], and major depressive disorder. The composition alters with aging and also tends to change in scale [146,147,148]. Understanding their precise roles in the disease pathogenesis of these disorders can help us develop new clinical interventions that range from developing novel diagnostic biomarkers to discovering new therapeutic approaches with increased specificity and better outcomes.

The role of the microbiome to trigger the immune system into alternative states has recently become a widely studied topic. Strong associations between changes in the microbiome and different autoimmune and inflammatory conditions have been discovered. Under certain circumstances, the antibodies produced against the antigens associated with the microbiome can act as autoantibodies, thus recognizing the “self” as a foreign invader and destroying it. This has put the autoimmune disease pathology under the radar of microbiome research. Rheumatoid arthritis (RA) is one such example [149,150,151,152]. Rat model studies have shown that the presence or absence of microbes can change the course of the disease [153,154]. Another recent study conducted on transgenic mice has shown differences in microbial composition that correlated with disease susceptibility or otherwise through the altered permeability of mucosa along with variations in the T helper 17 (TH17) transcriptomic profile, the determinant of the host response to microbes that also controls inflammation and tissue destruction [155]. These insights lay the foundation whereupon human clinical investigation can be based and explored.

Another mechanism postulated for the apparent link between gut microbes and autoimmunity that leads to rheumatoid arthritis proposes the idea that joint inflammation may be a consequence of toxic metabolites resulting from imbalances in gut microbes. This view is based on a causal link between the abundance of various microbial species and strains associated with increased risk of certain diseases. Gut *Prevotella*, a Gram-negative anaerobe, has been associated with many inflammatory and autoimmune conditions [156,157,158,159]. Of course, *“*pan-microbiome*”* studies are needed to provide a deeper understanding of the role of the human microbiome in RA pathogenesis, but it is sufficient to highlight the significance of microbes’ involvement in the autoimmune process. Another mentionable example where postulation of microbes involvement has given hope for understanding pathogenesis and treatment of periodontitis. By definition, periodontitis is a prevalent inflammatory condition of gums due to bacterial infection that, through unknown mechanisms, leads to complications. Dutzan and colleagues have demonstrated the implication of resident memory TH17 cells in human periodontitis, similar to rheumatoid arthritis [160]. The role of the microbiome in triggering inflammation is also evidenced by its association with the risk and pathogenesis of various inflammatory skin conditions. Emerging evidence also implicates microbiome in inflammatory conditions of the skin [161,162], such as atopic dermatitis, psoriasis, acne, and hidradenitis suppurativa [56,163,164,165,166,167].

A strong association between the incidence of inflammatory bowel diseases and an important controller of microbial infiltration, the interleukin-23 receptor (IL23R) gene, has been reported [165]. In addition, low plasma levels of vitamin D [168,169,170] and alterations in distribution and expression of its receptors in the gut [171,172] have also been designated as risk factors of inflammatory bowel disease [IBD]. In addition, the unavailability of anti-inflammatory short-chain fatty acids (SCFAs) [173,174] due to reduced diversity of the gut microbiome can be a risk factor for IBD [175,176,177,178]. This may occur as a consequence of various, more specific functional changes. In supporting this notion, the increased levels of Enterobacteriaceae may occur due to differences in its ability to tolerate the inflammation-associated redox stress [176]. More generalist or opportunistic Enterobacteriaceae can out-compete the short-chain fatty acids (SCFA)-producing *Clostridia*, leading to decreased microbial production of SCFAs that can eventually lead to a self-reinforcing pro-inflammatory state. These insights can prove beneficial for future research.

The role of the microbiome in these inflammatory disorders suggests that the microbiome serves as an essential regulator of inflammation and tends to control the pathogenesis of various inflammatory diseases. So now, let us look at the interaction of some critical elements of inflammation with the microbiome that can lead us to the eventual subject of neuroinflammation.

### Microbiome and Altered Cytokines

As we have reviewed the evidence in the previous section, the microbiome interacts with the host’s immune system and tends to modulate some of its activities and functions. In doing so, the microbiome also alters the cytokines through the microbial metabolites by dictating the dynamics of pro- and anti-inflammatory T cells. This leads to alterations in the Toll-like receptor (TLR) signaling [179,180,181], leading to the induction and proliferation of Treg cells and TH17 cells [179] and dampening of the intestinal epithelial cells (IEC) responses [180]. The proposed role of the microbiome to regulate inflammatory cytokines concentrations renders the microbiome a vital player in determining the dynamics of the immune response.

## 5. Microbiome and Vagal Control of Inflammation

The vagus nerve innervates all or most of the gastrointestinal tract and is another connection between the gut and brain [182,183]. Bacteria flora in the gut produces neuroactive compounds that include γ-aminobutyric acid (GABA), serotonin, dopamine, and acetylcholine (ACh) [183]. These compounds activate 5-HT3 receptors [24] that lead to recognition of bacterial products such as bacterial lipopolysaccharides (LPS) and short-chain fatty acids SCFAs [184]. In addition, microbiota may also maneuver vagus nerve excitability by altering the concentration of hormones such as ghrelin and orexin, thus deciding the eventual fate of a neurological output [185]. The TLRs are expressed on the vagus nerve and sensing bacterial products, resulting in the activation of nodose ganglia. EECs may also release serotonin on the nerve endings to alter the vagal nerve fibers [186]. In turn, when stimulated, these afferents result in reflex activation of efferents, giving anti- or pr-inflammatory privileges to the microbiome [187]. The inflammatory cascades may eventually result in alterations in the flora itself [188], which is why this interaction route is not physiological. Most of these processes are activated in situations such as endotoxemia which leads to drastic changes in the membrane permeability of the gut and leakage of bacterial products [189]. These processes may also be activated by stress, resulting in an increase in the permeability of the intestinal membrane [190,191,192] and the release of bacterial products and modifications in the gut flora in the long run [193].

## 6. Microbiome and Neuroinflammation

The human microbiome affects the immune response, definitely maneuvers inflammatory response, and seems to have the upper hand in regulating inflammatory activity inside the brain, the neuroinflammation [194], becoming a vital candidate for regulating the physiology of the brain, the behavior, and cognition. The patients of irritable bowel syndrome [IBS] have been found to benefit from probiotic *Bifidobacterium longum NCC3001* (BL), both in IBS symptoms as well as its depressive component [195]. The probiotic’s positive effects have been associated with the changes in the brain’s activation patterns and reduction in limbic activity.

This strong correlation has given birth to the term gut–brain axis, which attempts to explain the mutual impact of gut and brain on each other’s function. Alzheimer’s disease [AD] can be discussed as an example here. Firstly, the association of a fungal infection and AD afflicted brains has been reported [196]. Secondly, preliminary studies on the APP/PS1 mouse model of AD shed light on the links between the disease phenotype and alterations in the gut flora [197]. The results have shown that changes in flora may precede changes in inflammatory mediators and that might accelerate the inflammatory consequences of AD. Supporting this hypothesis, Lukiw W. J. has concluded in a review that LPS exposure of human primary brain cells can induce neuroinflammatory consequences through activation of NFκB (p50/p65) complex [198]. The gut bacteria can also activate neuroinflammation through the production of amyloids [199]. The strongest link can be established from the studies that link reinforcing the normal or beneficial flora and seeing changes in the progression of disease susceptibility or the disease itself. Evidence supporting the role of neuroinflammation is that the dietary changes that affect the microbiome can also affect neuroinflammation. Studies on *APOE4* transgenic (E4FAD) mice have shown that dietary changes that enhance the beneficial microbiota and decrease harmful microbiota can preserve systemic metabolic function and reduce neuroinflammation [200]. This implies a quintessential causal link between the two. However, the bacterial load needs to be considered as it can affect the eventual impact of inflammatory processes. It has been found that antibiotics-induced reduction in bacterial load in *C57BL/6J* female mice was a factor that saved the mice from alcohol-induced neuroinflammation. Lowe et al. [201] have reported that alcohol increases the expression of proinflammatory cytokines in the brain and intestine, which is reversible by the use of antibiotics. A similar argument comes from studies conducted on the subject of Gulf War Illness. Gulf War Illness (GWI) clinically presents with various neurological abnormalities, chronic fatigue, and gastrointestinal disturbances [202]. Rodent model studies of GWI have shown that chemical exposure can cause significant alterations of bacteria in the gut, causing a leaky gut [203]. The leaky gut leads to TLR4 activation and, consequently, neuroinflammation [204,205]. An altered gut–liver–brain axis may be a risk factor for hepatic encephalopathy (HE) irrespective of the precipitating cause [206,207]. After antibiotics administration, fecal microbial transplant (FMT) can potentially improve the outcomes in HE [208]. FMT, as the name indicates, restores the normal microbiota to the best possible state, which eventually leads to some normalization of the gut–brain communication about inflammation. Murine model studies have suggested that the FMT can attenuate the neuroinflammation by restoring the balance in the gut [209].

As discussed earlier in the section on aging, cognitive performance significantly declines with aging. Neuroinflammation is an important accused in this context. A critical insight in this regard has come from the gut microbiome-derived metabolite trimethylamine N-oxide (TMAO), which has been implicated in the age-related neurocognitive decline and has also been shown to modulate neuroinflammation in middle-aged and older human subjects as well as mice [210]. In contrast, practical strategies to restore microbiota balance and reduce dysbiosis through the consumption of butyrate and dietary soluble fiber lead to improvement in the parameters of neuroinflammation associated with aging in mice [211]. The microbial dysbiosis in the gut has a bidirectional relationship with the neuroinflammatory response after stroke. On the one hand, it is stated to control the post-stroke neuroinflammatory response [212]. It is also a consequence of the neuroinflammatory process at the same time as large stroke lesions alter the balance in the gut due to dysfunctional regulation, which can, in turn, affect the outcome of the stroke via immune-mediated mechanisms [213]. Animal model studies have shown that the dysbiotic microbiome following a stroke can induce proinflammatory T-cell polarization within the ischemic brain and the intestinal immune compartment [214]. The microbial dysbiosis thus controls the neuroinflammatory response after the stroke. Similarly, synucleinopathies, characterized by the aggregation of α-synuclein (αSyn), can cause motor dysfunction as observed in PD. Sampson et al., in one of the most cited articles on this subject, have reported that alterations in gut microbiota can promote motor deficits, microglia activation, and αSyn pathology, which was reversed by the use of antibiotics suggesting the role of postnatal signaling between the gut and the brain in disease modulation [215]. They further showed that the oral administration of specific microbial metabolites in the absence of germs could promote neuroinflammation and motor symptoms, which justifies the causal association of gut microbes with inflammatory processes in the brain. This relationship has also been highlighted in neurological symptoms following COVID-19 infections. The severe acute respiratory syndrome coronavirus 2 (SARS-CoV-2) infection has been proposed to play an essential role in increasing neuroinflammatory and neurodegenerative disorders such as AD and PD [216,217]. This newly reported virus can cause an imbalance of the gut microbiome, intestinal inflammation, and dysregulation of the gut–brain axis. However, more data from clinical studies aimed at estimating the risk of various CNS-related disorders in SARS-CoV-2 infected individuals are still needed to confirm this hypothesis. Together these findings suggest that the gut microbiome can play an essential role in the process of regulation of neuroinflammation. The precise mechanisms through which these microbes play this role are many and not fully understood. However, these studies do connect some dots that can help formulate future therapeutic strategies accordingly.

## 7. Microbiome, Microglia, and Neuroinflammation

The microglia are the innate sentinel immune cells in the CNS capable of detecting subtle changes in the associated nervous tissue [90]. Owing to their critical roles, they are of prime importance in neuroinflammation and serve as the producers of various molecular promoters and mediators of neuroinflammation [218]. Microglia constitute about 5–12% of the total brain cells [219]. They are involved in neuroinflammatory events of all origins [220]. In the CNS, microglia also participate in regulating neural patterning and synaptic pruning during development and throughout adulthood [221]. Upon activation, microglia release a wide variety of chemokines and cytokines along with the expression of numerous antigenic markers [222]. They also tend to regulate the production and release of various neurotransmitters and undergo extreme morphological changes in response to local and peripheral tissues [223,224,225]. Microglia respond by assuming an activated phenotype and become highly motile [225]. In this activated phenotype, microglia release pro-inflammatory [226] and anti-inflammatory cytokines to maintain homeostasis [227]. Chronic microglial activation can lead to irreversible neurodegeneration [228].

Despite residing in a highly secluded environment guarded by the BBB, microglia’s maturation and immunological function remain under the gastrointestinal microbiota’s influence. The diversity of gut microbiota has been associated with increased microglial populations; alterations to microglia morphology (with increased branching, longer processes, terminal points, and clubbing at synaptic boutons); defects in microglia maturation, differentiation, and activation state; and a compromised immune response to invading pathogens [229]. The reversal in the microglial phenotype was observed with recolonization of gut microbiota [229]. This relationship can be explained by proposing a role for the microbiota in microglia development and maturation. The absence of microbiota has been linked to arrest in the immature developmental stage of the microglia and displayed limited responses to viral infection and microbial antigens [230]. In germ-free mice, the BBB shows increased permeability towards macromolecules compared to conventionally raised animals, while restoration of pathogen-free gut flora seemed to re-establish tight junctions and decrease BBB permeability [231]. Aberrant changes to both microbiota and dysregulated microglial function have been linked to neurobehavioral, neurodevelopmental, and neurodegenerative disorders, including depression, anxiety, and autism spectrum disorder, in addition to AD and PD [232].

Soon after birth, primary seeding of microbes occurs in the gastrointestinal tract and elsewhere. The composition of this colonization is determined by the mode of delivery, feeding source, and exposure to the environmental elements [233]. It is important to establish the flora and facilitate the colonization because it determines brain development [234]. Microbes have been found to regulate HPA axis activity in mice [235].

How does dysbiosis alter the state of immune cells inside the brain? There are bits of pieces of information that may help shape up a theoretical explanation of this connection (Figure 2). The microglia are affected by the secretion of bacterial endotoxin lipopolysaccharide (LPS) from gut flora. LPS is a bacterial membrane component found in Gram-negative bacteria that acts as an endotoxin. It has been shown that LPS directly affects microglial activation and has been implicated in the pathogenesis of subsequent neurological disorders and behaviors. Intraperitoneal administration of LPS can increase serum cytokine levels of TNF-α, MCP-1, IL-1β, IL-6, and IL-10. Additionally, Hoogland et al. have demonstrated increased microglia activation after LPS administration and *E. coli* infection [236]. The effect of LPS on cytokines’ release essentially explains the connection between the endotoxin and its neurological effects. Based on this fact, LPS is frequently used to investigate the effects of inflammation on behavior in rodent models, as systemic injection of LPS can induce depressive-like behaviors in animals [237]. The LPS induced sickness behavior model, also called the depressive-like behavior model, is also predictively validated as blocking TLR4-LPS recognition can attenuate the sickness behavior observed after the LPS challenge [238]. In addition to stimulating the release of cytokines, LPS also causes activation of the IDO enzyme, which alters the state of microglia from nascent to active [239]. Collectively, these observations suggest pathways under which the causal role of the GMB in altering the neuroinflammatory equation and reiterate its strong position for due consideration in therapeutics.

## 8. The Microbiome and the HPA Axis

The HPA axis is the primary response system of the organism when faced with stress [240]. It is duly affected by microbiota, and the dynamics of this influence are determined by bacterial strain and availability of substrates that it metabolizes and host factors such as age and gender [241]. Depending on the strain, this interaction employs a diverse set of mechanisms that result in changes in the HPA response towards stress [242,243,244,245,246]. Given the direct impact of HPA axis alterations on the inflammatory response, the microbiome constitutes a trilateral stress response system with the microbiome, neuroendocrine, and neuroinflammatory components [247]. For example, the STAT5 signaling pathway that connects microbiota and neuroendocrine and neuroinflammatory pathways have the capacity to regulate many cytokines [248], such as IL-3 [249]. Its role has been recognized in neuroprotection, apoptosis, proliferation, and differentiation, as well as hematopoiesis and immunoregulation [250]. The susceptibility of the STAT5 signaling pathway to changes in the microbiome composition [251,252,253] means alterations in microbiome composition affect the neuroinflammatory process and related disorders. Another mechanism of interest to overview here is the de novo synthesis of corticosteroids in the intestine, which is directly influenced by intestinal microbiota [254,255]. These glucocorticoids, along with the CRH and glucocorticoids from the HPA axis, can influence microglial function [256]. Due to its negative feedback nature, glucocorticoids are of crucial importance as higher corticosterone levels suppress macrophage activation (anti-inflammatory).

In contrast, lower corticosterone levels enhance the production of pro-inflammatory factors [257]. Glucocorticoids have been found to alter the inflammatory mediators’ expression, thus acting as an anti-inflammatory in nature [218]. Thus, microglia can be influenced by intestinal glucocorticoids and metabolic cues from the microbiota and the HPA axis, and as such, regulation of microglial functions remains susceptible to changes in the human microbiome.

### Microbiome Alterations and Neuroinflammation in Autism Spectrum Disorder

Autism spectrum disorder (ASD) is diagnosed in children favoring the male gender ratio prevalence [258]. Autism spectrum disorder (ASD), occurring in 14.6 per 1000 children, with a 4:1 male to female ratio, is a complex neurodevelopmental disorder characterized by a deficit in communication and interactions, minimal interests, and repetitive behavior [259]. Multiple genetic and environmental factors are hypothesized to contribute to ASD [258]. Biologically, it is defined by reduced neuronal connectivity, immune dysfunction, microbiome dysregulation, mitochondrial abnormalities, microglial abnormalities, and gastrointestinal distress [260,261]. Up to 90% of ASD children are diagnosed with gastrointestinal distress, which has become a critical player in the disorder and the main topic in recent autism research [262]. The gut microbiota appears to be implicated in the gastrointestinal distress symptoms in ASD [263] through the signaling of the gut–brain axis [264,265]. In addition to the usual stimulation of the immune system by releasing proinflammatory cytokines or through the production of SCFAs, altered gut permeability also plays an important [266,267] ASD pathogenesis.

Although the detailed mechanism of the gut–brain axis is still to be unraveled, studies in the last couple of decades on the scope of alterations in GMB in the morbidity of ASD have been extensively reviewed and offer vital information on the subject [268,269,270,271,272]. Different studies have found distinct species of microbiota related to neurodevelopmental disorders such as more *Rothia* species [273], less *Moraxella*, *Megasphaera*, *Neisseria*, and *Gemella sps* [273], and reduced *Firmicutes* but elevated *Acidobacteria* percentage in autistic compared to healthy subjects (*p* < 0.05) [274]. In a systematic review on the subject, Wang et al. investigated the GMB in 92 ASD children by shotgun metagenomic sequencing and mass spectrometry. They indicated the presence of altered glutamate metabolite due to decline in 2-keto-glutaramic acid associated with the altered microbiota composition, i.e., reduced levels of *Bacteroides vulgatus* and increased levels of *Clostridium botulinum* and *Eggerthella*
*lenta* [275].

Several reports calculate that every human gut is inhabited by 500 to 1000 varieties of bacterial species [276,277]. The relationship between autism and individual bacterial species has been explored, and a wide and complex variety of alterations has been found between different individual components of the microbiome [278,279,280,281,282,283,284]. Therefore, it would be wise to focus on their mechanism of pathogenesis instead of namedropping. The mainstay of gut microbes’ interaction with the immune system is either through the release of proinflammatory cytokines or the production of short-chain fatty acids (SCFAs) [58]. The translocation of the gut bacteria provokes the release of the proinflammatory cytokine from the intestine into the mesenteric lymphoid tissue, leading to increased gut permeability. This cytokine concentration change is eventually reflected inside the brain tissue and is linked to ASD pathogenesis [285]. Inflammatory markers, including TNF, IL-6, and monocyte chemotactic protein 1 (MCP-1, a chemoattractant for mast cells), were identified in the brains and cerebrospinal fluid in many ASD patients [79]. Mast cells and microglial interactions are essential in neuroinflammatory diseases as mast cells, pro- and anti-inflammatory mediators, and microglia are quintessential components of the neuroinflammatory process [286].

Along with vasoactive neuropeptide neurotensin (NT) receptors in Broca’s area, mast cells’ abundance in diencephalon may directly explain the linguistic and behavioral alterations seen in ASD [270,287]. Furthermore, mast cells have an essential role in modulating immunity via the production of both pro and anti-inflammatory mediators [282]. Therefore, in ASD children, it is highly likely that mast cell activation is caused by non-allergic triggers leading to allergic-like reactions [264]. Moreover, they are abundantly located in the brain’s diencephalon, which regulates behavior [264]. Additionally, mast cells and microglial interactions are essential in neuroinflammatory diseases [288,289]. Abnormal microglia growth and activation have been reported in ASD patients [290,291].

Another key player in inflammation is the vasoactive neuropeptide neurotensin (NT), found in both the brain and the gut [292]. NT receptors are highly concentrated in the Broca area, which regulates language known to be lost in many children with ASD [284]. The significantly high serum levels of NT in ASD children trigger mitochondrial DNA (mtDNA) secretion, which stimulates immune cells, including mast cells [293], leading to auto-inflammation [293]. Additionally, high serum levels of mtDNA in ASD children cause neuronal degeneration and alter behavior [294]. Stimulation of mast cells accrues when NT and CRH are released under stressful conditions, leading to increased vascular permeability of the BBB, its disruption, and inflammation [295,296]. Additionally, CRH receptor-1 (CRHR-1) expression is increased by NT, and when activated by CRH, it leads to the allergic stimulation of human mast cells [297,298]. This is compounded by the activation of the stress axis mediated by inflammatory cytokines. Studies suggest that the HPA axis functioning in persons with ASD is significantly diverse, with an overall sluggishness in response to cortisol responding to physiological or physical manipulation [299]. Moreover, individuals with ASD respond differently to stress by showing either a hyper-responsive HPA axis to different natural stimuli and social conditions or a hypo-responsive HAP axis to stressors, which is inactivated by evaluative threat [300]. Other mentionable components of the inflammatory impact of microbiota revolve around *clostridia* species predominant in ASD subjects both in stool samples [301] and intestinal biopsy studies [280]. These species orchestrate inhibition of dopamine-β hydroxylase, leading to increased production of reactive oxygen species resulting in inflammation [265]. The same species has also been found to have a correlational relationship with the tryptophan homeostasis and proinflammatory cytokines IL6, IL1, IL17A, and interferon-γ in ASD subjects [302,303], being identified as a potential link between intestinal microbiota, serotonin, and tryptophan secretion, and inflammation. Some of this evidence has been translated into practice as well. Oral intake of specific strains of bacteria [304], as well as fecal microbiota transplant (Kang et al., 2019), has yielded encouraging results in alleviating behavioral problems [265] and correcting inflammatory alterations [304,305].

Additionally, proinflammatory cytokines release such as IL-1 and IL-6, in particular, can cross the BBB and activate the HPA axis. The activation of the HPA axis can lead to cortisol release, which triggers the stress system [265]. Studies suggest that the HPA axis functioning in persons with ASD is significantly diverse, with an overall sluggishness in response to cortisol responding to physiological or physical manipulation [298]. Moreover, individuals with ASD respond differently to stress by showing either a hyper-responsive HPA axis to different natural stimuli and social conditions or showing a hypo-responsive HAP axis to stressors, which makes it inactivated by evaluative threat [298].

The gut microbiota and its composition have a strategic role and a deep connection with ASD. GI problems in ASD patients have been associated explicitly with *Clostridium (Lachno-clostridium) bolteae* [306]. Several studies have pointed to the increased colonized varieties of *clostridia* species in the intestinal tract in ASD patients [289,307]. In fecal samples isolated from ASD children, nine *clostridium* species were isolated that were not found in the predominant fecal microflora of healthy controls [277]. Moreover, three species were found only in healthy control fecal samples [277]. Additionally, beta-2 toxin-producing *Clostridium perfringens* counts were significantly higher in stool specimens of autistic children with gastrointestinal abnormalities [308].

Urine samples of autistic children had elevated amounts of glyphosate, a widely used herbicide that can reduce the beneficial bacteria in the gut microbiota [32,280]. Insensitive to glyphosate, *Clostridia* bacteria increases following glyphosate exposure, evidenced by high levels of *Clostridia* metabolic compounds 4-cresol, 3-[3-hydroxyphenyl]-3-hydroxy propionic acid (HPHPA), and 4-hydroxyphenyl acetic acid (4-HPA) in the urine samples, which are associated with autism [309]. These compounds inhibit dopamine-β hydroxylase, an enzyme that converts dopamine to norepinephrine in the brain and the sympathetic nervous system [310]. This leads to an increased amount of dopamine, which leads to the production of reactive oxygen species that damages Krebs cycle enzymes, the mitochondria, and neurofibrils [311].

While most studies have evaluated intestinal microbiome composition in ASD children based on stool samples, a study used intestinal biopsies to investigate the microbiota and gene expression in ASD children with gastrointestinal disease [312]. The study reported a deficiency in disaccharides and hexose transporters genes in ASD children, which was associated with lower levels of the intestinal transcription factor caudal type homeobox 2 (CDX2) [312], demonstrating deficiency in the primary pathway in carbohydrate digestion and transport in enterocytes [312]. Moreover, the microbiota composition in ASD children had increased *Clostridiales*, specifically *Lachnospiraceaea* and *Rumino-coccaceae*, compared to neurotypical children [312]. In another study, ASD children with functional gastrointestinal disorders (FGIDs) had an increase in mucosa-associated *Clostridiales* and lower levels of *Dorea* and *Blautia*, and *Sutterella* compared to controls (38). Furthermore, tryptophan homeostasis and proinflammatory cytokines IL6, IL1, IL17A, and interferon-γ were found to correlate significantly with multiple *Clostridium* species associated with ASD, which can alter neuroimmune signaling and could identify new potential links between intestinal microbiota, serotonin, and tryptophan secretion and inflammation [302].

Studies conducted on murine have tried to elucidate the critical links between ASD development, the microbiota, and environmental risk factors [313]. One study investigated the effect of maternal infection as a risk factor of ASD in the maternal immune activation (MIA) mouse model of maternal inflammation [314]. The yielded offspring, which display A SD-like behaviors, had altered immune profiles and function, a systemic deficit in regulatory T cells, increased IL-6 and IL-17 produced by CD4(+) T cells, and elevated levels of peripheral Gr-1(+) cells [314]. In a second study conducted on MIA mice, the offspring had gastrointestinal symptoms similar to human ASD resembled in altered intestinal integrity and dysbiotic gut microbiota composition due to changes in operational taxonomic units of *Clostridia* and *Bacteroidia* [315]. Moreover, probiotic treatment with human commensal *Bacteroides fragilis* of MIA offspring corrected intestinal permeability and tight junction alterations, restored increased IL-6 mRNA and protein expression, and inhibited specific changes in microbes of the *Lachnospiraceae* family and unclassified *Bacteriodales* [315].

Since there are no effective treatments to cure autism to date, several therapeutic approaches have been targeting the gut microbiome and the gut–brain axis via multiple approaches considering their central role in autism. One of the therapeutic interventions uses probiotics for children with ASD by using a single strain or a mixture of multiple strains such as *Bifidobacterium* spp. and *Lactobacillus* spp., which may stimulate immunity, reduce toxins and pathogens, and increase antioxidants and vitamins [303]. Moreover, it stimulates oxytocin production and positively influences ASD social behavior [265]. In contrast to probiotic treatment, fecal microbiota transplantation is another therapeutic approach that focuses on transferring several hundred bacterial strains from a donor to a recipient, which showed efficiency in recurrent *Clostridium difficile* infection, and in enhancing ASD behavioral symptoms and gastrointestinal problems [265].

Since the dream of configuration and restoration of the ideal composition of individual microbiome remains elusive, there are no effective treatments to cure autism to report. However, several therapeutic approaches have been proposed targeting the gut microbiome and the gut–brain axis via multiple approaches considering their central role in autism. One of the therapeutic interventions uses probiotics for children with ASD by using a single strain or a mixture of multiple strains such as *Bifidobacterium* spp. and *Lactobacillus* spp., which may stimulate immunity, reduce toxins and pathogens, and increase antioxidants and vitamins. Moreover, it stimulates oxytocin production and positively influences ASD social behavior. In contrast to probiotic treatment, fecal microbiota transplantation is another therapeutic approach that focuses on transferring several hundred bacterial strains from a donor to a recipient, which showed efficiency in recurrent *Clostridium difficile* infection, and in enhancing ASD behavioral symptoms and gastrointestinal problems. Further investigations remained needed to fully elucidate the molecular mechanisms behind the connection of the gut–brain axis, the microbiota, and inflammation in ASD.

## 9. Microbiome, Behavior, Depression, and Personality

Given the diverse role of the microbiome in neurological function in health and disease reviewed here, it is not wrong to consider the microbiome as a critical determinant of an individual’s behavior. The crosstalk between the brain and the gut can be moderated at four different levels, including (i) the immune system, (ii) the parasympathetic nervous system, (iii) via delivery of the neurotransmitter and neuroactive metabolites produced in the gut via the circulatory system (iv), and via the gut neuroendocrine system. For example, an estimated 90% of all serotonin is produced by enterochromaffin cells in the gastrointestinal tract and is influenced by the gut’s microbiome [46]. Although the circulating serotonin is primarily metabolized by the liver and cannot cross the BBB, it can affect the vagus nerve activity and BBB permeability. Alam et al. have reviewed the role of bioactive nutrients and gut microbiota in altering the DNA methylation and histone signatures and their impact on the gut–brain axis and found out that the gut microbiota can regulate the inflammatory cytokines and affect the epigenome by the generation of SCFAs, nutrient absorption, and vitamin synthesis [316].

Alterations in commensal intestinal microbiota or pathogenic bacteria result in subtle changes in behavior [251]. The results from mice raised in germ-free environments, conventionally housed animals, and mice treated with probiotics and/or antibiotics or infected with pathogenic bacteria all indicate that rodent behavioral responses are impacted when the bacterial status of the gut is manipulated [317]. Germ-free housing, as well as antibiotic treatment, reportedly reduces anxiety-like behavior in the light/dark test (L/D), elevated plus maze (EPM), and the open field (OF) [318]. However, these studies have shown a critical window during development whereby the CNS wiring related to stress-like behavior is influenced by microbiota [187]. Probiotics reportedly also influence anxiety-like and depressive-like behaviors [319]. While a history of infection and gut inflammation increases anxiety-like behavior [320], treatment with *L. rhamnosus* decreased anxiety-like and depressive-like behaviors in healthy male *Balb/C* mice [321]. In humans, treatment with probiotics *Lactobacillus helveticus R0052* and *Bifidobacterium longum R0175* leads to decreased score in Hospital Anxiety and Depression Scale (HADS) score [322]. Healthy human prebiotic treatment with *fructooligosaccharides* or *galactooligosaccharides* showed diminished salivary cortisol and increased positive versus negative vigilance [323]. Reduced cognitive reactivity to sad mood with aggressive thoughts was reported in the patients of mood disorder after probiotic treatment [324].

In clinical settings, the microbiome has become a subject of interest for the treatment of refractory depression. Of course, the individual components examined so far in this review are already implicated in the pathogenesis of depressive disorder. After examining evidence related to these components, we now review the available data directly related to the clinical setting of treatment-resistant depressive disorder. TRD primarily refers to difficult-to-treat or refractory depression. There is a great deal of inconsistency in the definition of TRD. The most common definition of TRD is “the depression that is non-response to two or more adequate courses of consecutive antidepressant treatment during a single major depressive episode (MDE) [325]. TRD affects almost 20–30% of all individuals suffering from MDD and contributes considerably to the overall disease burden [326]. TRD is associated with higher rates of relapse, poorer patient health-related quality of life (HRQoL), and increased mortality rate as compared to non-TRD within one year of remission [327]. Individuals with TRD usually have minimal therapeutic options. Fluoxetine/olanzapine is currently the only explicitly approved therapy for TRD in the United States (US) [328].

Over the past few years, increasing evidence has suggested a strong link between the microbiome composition and the development of mental disorders, including depression. The fecal microbiome composition in patients diagnosed with MDD has been found to be different compared to healthy controls [329,330,331] (reviewed by [332,333,334,335]) with increased Proteobacteria, Bacteroidetes, and Actinobacteria and fewer Firmicutes in MDD patients [329].

As we have discussed in the earlier sections, the resident gut microbes can support the brain and the immune system. The gut microbiome composition strongly influences the HPA axis. The male mice raised in a germ-free environment have an over-reactive HPA axis that leads to increased production of corticosterone and adrenocorticotropic hormone (ACTH) in response to a stressful stimulus [336]. An epigenetic study has highlighted that the expression profiles in various regions of the brain of germ-free mice also differ from controls and serve as the contributors for the differences observed in the HPA axis activity [187]. The synthesis and release of neurotransmitters were also found to be different between the two groups. Collectively, it all highlights the significant changes in the brain of murine models raised in a germ-free environment and advocates the perpetual impact of the microbiome on homeostasis [337,338,339].

The use of probiotics has been shown to have mixed effects on anxiety and depression; however, various studies have shown that the use of some probiotics can improve the immune response, function of the intestinal barrier, stress response, and ultimately mental health [340,341]. *Lactobacillus (L)* spp. probiotics have been shown to ameliorate the over-reactive HPA axis in mice facing stress due to maternal separation. Administering *Lactobacillus rhamnosus* can significantly reduce the corticosterone levels in the mice subjected to stress-induced hypothermia or the elevated plus-maze test along with region-dependent alterations in γ-aminobutyric acid (GABA) receptor subunit expression in the brain. Pre-treatment with the probiotics *L. helveticus* and *B. infantis* has been shown to reduce stress response and maintain hypothalamic plasticity and hippocampal neurogenesis. A study conducted on healthy medical students orally taking *L. casei* showed that the probiotic significantly improves the academic stress-induced increase in cortisol levels.

Probiotics and prebiotics can also decrease inflammation, thus becoming important determinants of the response to the anti-depressants. Of course, the complexity and peculiarity of the human gut microbiome make it difficult to design an intervention that restores “the ideal flora” and heals every disease. Clinical trials and studies are, thus, just breaking the ice in finding therapeutics based on the restoration of beneficial bacterial populations. Probiotic *Lactobacillus farciminis* administration has been found to improve intestinal integrity, reduce its permeability in mice, and reduce partial restraint stress-induced hyperpermeability of the colon, thus preventing the endotoxins from entering the circulatory system and inhibiting endotoxemia [342]. Additionally, *L. farciminis*, *L. helveticus*, *Bifidobacterium longum*, and *Lactobacillus plantarum* have also been reported to substantially affect intestinal integrity [343]. Treatment with a mix of *L. helveticus* and *B. longum* probiotics in healthy human volunteers for one month suppressed psychological distress [344]. Another study conducted on IBS patients showed that high prebiotic trans-GOS intake for 12 weeks could improve anxiety scores and subjective global assessment scores [345]. Intake of probiotic-containing milk daily for three weeks in volunteers in the bottom third on the depressed/elated dimension reportedly showed mood improvement compared to the placebo [319]. Cognitive reactivity to sad mood was also shown to be reduced in healthy human subjects receiving a multispecies probiotic [324]. The probiotic treatment has reduced the risk of side effects as compared to the conventional drug regimens [346]. Along with microbial-associated molecular patterns (MAMPs), these factors can affect immune system activity, intestinal barrier integrity, and vagal nerve stimulation to improve depression and anxiety [347]. Microbiome-derived SCFAs can promote gut homeostasis in non-inflammatory conditions and promote serotonin production in enterochromaffin cells [348]. These fatty acids are also capable of bolstering barrier integrity in intestinal epithelial colon cells, thereby preventing endotoxemia from entering circulation [180]. SCFAs can also regulate the development of a variety of immune cells, including dendritic cells, regulatory T cells (Tregs), and microglia [349]. These fatty acids can also downregulate various pro-inflammatory mediators. The microbiome-derived Trycats are aryl hydrocarbon receptors (AHRs) agonists, ligand-dependent transcription factors expressed on intestinal epithelial cells and the immune cells [350]. Microbe-derived Trycats can bind to AHR, expressed on innate lymphoid cells, induce IL-22 release, and promote antimicrobial peptide production and pathogen defense [351]. Trycats can also promote the differentiation of IL-10-producing T-Regs cells in place of pro-inflammatory Th17 cells [352]. These molecules are also capable of limiting neuroinflammation by binding to AHRs on astrocytes. Altogether, the probiotics intake can potentially regulate the stress response and decrease the intestinal permeability in both murine and humans. In addition, the gut microbiome can alter both stress and inflammation, which play a crucial role in the modulation of brain function and thus exert antidepressant and anxiolytic effects. Considerable evidence suggests that these improvements can lead to decreased anxiety- and depression-like behaviors in rodents.

The above-stated evidence suggests that the human microbiome study can help discover new drug candidates for depression that might help patients with TRD. Probiotics and the microbes that improve intestinal integrity, reduce inflammation, and positively impact brain function can treat depression that is not responsive to any of the current treatment regimes. Fecal microbiota transplants (FMT) have shown promising results for treating depression and anxiety [353]. In addition, FMTs have shown promise in improving neurological diseases [354]. Another study supporting the role of FMTs in improving mental health is that the germ-free mice receiving FMT from patients with depression exhibited more depressive-like behaviors than GF mice that received FMT from healthy control individuals [353]. Currently, many questions need to be answered yet. However, microbial transplants and probiotics to promote the growth of CNS-friendly microbes are an up-and-coming field that can help many TRD patients.

While long-term effects of dysbiosis in the critical development window have been documented for many diseases [355], not much data is available for behavioral or personality-related subjects, and therefore, only hypotheses can be made. Early life composition determines the adult distribution of the microbiome in an individual [233]. However, it is an ongoing debate if primary seeding of the microbiome in neonates plays a role in shaping an individual’s long-term mental health and personality. The inoculation of vaginal swabs is being advocated and practiced in parts of the world for babies delivered through cesarean section [356,357]. For centuries, however, cultural rituals have been performed that have presumably benefited newborns in establishing a healthy flora in their gut. In the Punjab area of Pakistan and India, a peculiar variation of an otherwise worldwide known ritual exists for newborn children that merit a mention in this discussion. The inoculation ritual is commonly referred to as the administration of ghutti. An elder member of the family, usually the most likable person in terms of temperament and personality, is asked to administer first food to the newborn. The ritual goes like this: the elder takes something sweet and soft, e.g., a date, then chews it in their own mouth to make it even softer and subsequently inoculates the chewed soft and sweet date onto the upper palate of the new-born with their thumb or finger of the right hand. The act of chewing in one’s own mouth is peculiar to Punjab and may possibly be the oldest version of the act. As far as the discussion among the authors yielded, the ritual exists in Arab culture as well with the name of tahnik, where first food is administered by a chosen one and is believed to bless the child with their excellent personality and habits. Of course, there is no empirical evidence so far that there is any correlation between the gut microbiome composition and the oral microbiome of the person who is doing the ritual, and the practice has largely been given up probably for concerns about hygiene and appropriateness for the baby. It is believed that the agreeable elder will transmit good features of his or her personality to the newborn by this inoculation. The authors have wondered how the ancient cultures connected the desired personality features of the elder person being transferred to the newborn with food. It may, in fact, not be the food but the microbiome that can make a difference. The authors, being from different cultures, discussed it at length and postulate it might be this first inoculation of the microbiome that determines the relative composition of the gut microbiome in adult lives and be responsible for the reflection of temperamental and personality features and/or behavior of the baby. Elders in Punjab often ask for a reminder about the administrator of ghutti when they observe something extraordinary in their subsequent generations. A study on microbiome composition found a strong correlation between enterotypes and temperament and personality in the adult life in healthy Korean adults, which was attributed to biochemical signature determined by early colonizers of the gut [358]. Another notable Korean study found that disbalance in gut microbiota and neuroticism may have a close association, and this relationship can be exploited for potential microbiome-based interventions [359]. Many other studies have validated similar ideas [360,361,362,363,364,365,366,367,368]. The subject of primary inoculation and cultural practices needs more basic research for the proper building of the evidence, but the first bricks are laid.

## 10. Causative Analogy and the Possibility of Restoring the Lost Balance

As with many theories underlying biological disorders, the causative analogy between alterations in microbiome composition and a particular disease state is as tricky as it looks. The primary reasons are the diversity of the microbiome and the limitations of data required for establishing a causal link. The microbiome is so diverse that each of the seven billion humans on earth may have a particular composition that no one else has. The phenomenological advances have been promising, and the microbiome does seem to have the potential of becoming a discipline with therapeutic potential [369]. It appears as if microbiome signature research will have to go through computational models if it ever wants to make it to therapeutics. The data has led to many clinical trials [344,370,371]. The studies based on forward genetic techniques are encouraging but inductive in nature, which means they project specific results on the whole. Inductive inferences have long haunted biological research, and a diverse field such as microbiome makes inductive reasoning more of a problem for scientists. Of course, a philosophical discussion of the subject is not in the scope of the review. However, computational models can help solve this puzzle. Individual bits of information can be computationally put together for building up the hypotheses and interventions. Direct interventions, e.g., using specific species of good bacteria to restore the ideal composition, have led to failure in highly expensive clinical trials [372], although it was based on some extraordinary results obtained in earlier phases [373]. Instead, fecal microbial transplant offers a whole other level of diverse and balanced composition that can re-establish the lost equilibrium [374]. An auto-transplant of a previously preserved fecal sample can prove even better during recovery from a potential state of disease that has altered the composition of a person’s intestinal microbiome (Figure 3).

## 11. Microbiome Project in South Korea

In addition to the previously cited Korean studies linking microbiome with different aspects of neurocognitive aspects of human personality, two microbiome-related projects carried out in Korea merit to be mentioned here. Carried out between 2010 and 2015, the Korean Microbiome Diversity Using Korean Twin Cohort Project and Fecal Microbiome Projects have added great value to the existing knowledge in this area of immense therapeutic interest. This includes newer species of microbiota found in the feces of the animals, as well as building a database of the fecal microbes that, in the future, prove beneficial in redefining the actual value of microbiome in therapeutics.

## 12. Conclusions and Future Prospects

In summary, microbiome is a crucial part of human physiology that plays a vital role in various critical biological pathways. Perturbations in the normal microbiota can lead to various diseases. The microbiome has the ability to regulate the immune system, maintain intestinal and BBB integrity, modulate parasympathetic nervous system, brain function, and neuroinflammation. Understanding the regulatory mechanisms that govern the microbiome–CNS interaction can help understand the pathological mechanisms that underlie various neurological disorders, including depression and anxiety. The results from the various animal models and human studies have shown that probiotics and microbial transplants can positively influence anxiety, stress response, and depression and can thus help patients suffering from various neurological diseases. However, the mechanisms that influence the role of microbiota in the pathogenesis of these diseases need to be further elucidated. The understanding of these mechanisms can help in the identification of the microbial organisms that can have a significant impact on depression. The studies conducted to date are minimal, and none has been conducted exploring the role of probiotics and microbial transplants in TRD patients. The design and conduct of such studies are a promising field that can be explored to help patients that are not responding to any of the current treatment regimes.

## Figures and Tables

**Figure 1 microorganisms-10-00705-f001:**
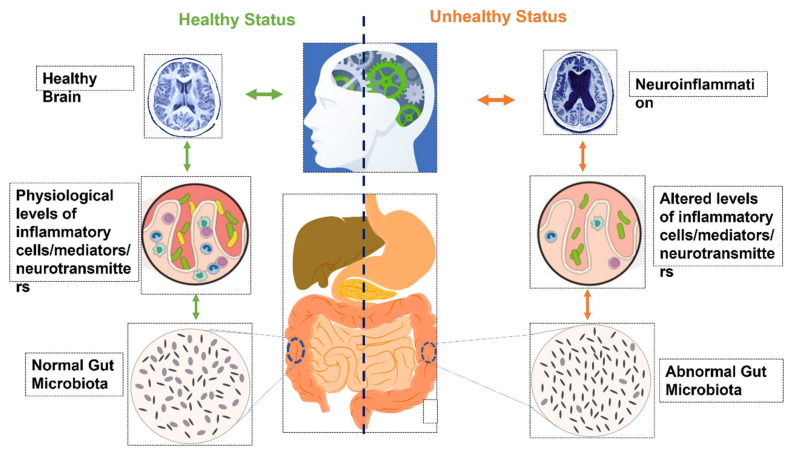
Illustration of the impact of the gut microbiome on neuroinflammation.

**Figure 2 microorganisms-10-00705-f002:**
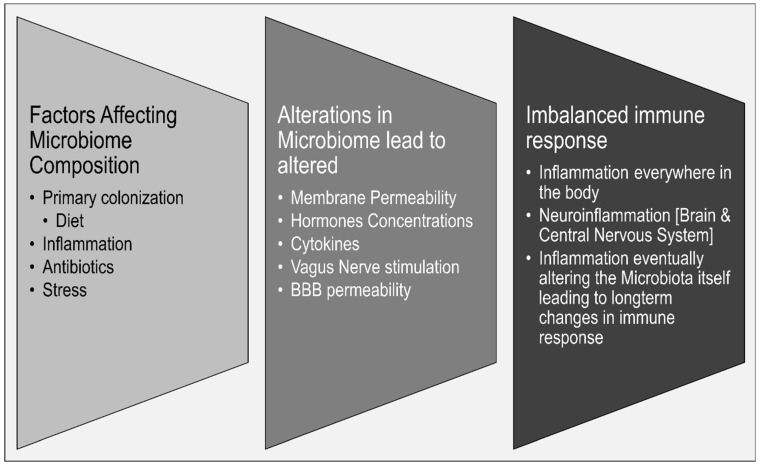
A summary of various factors that connect microbiome and neuroinflammation.

**Figure 3 microorganisms-10-00705-f003:**
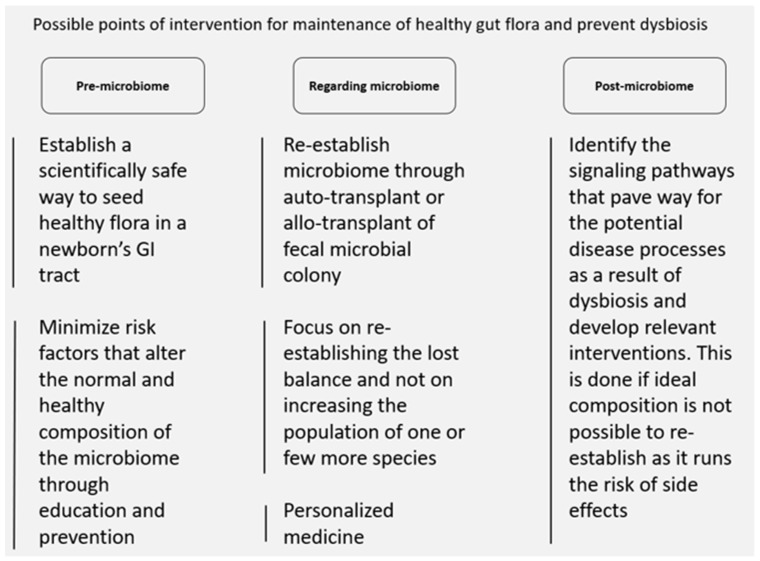
Proposed interventions that may help maintain microbiome, leading to homeostasis in the brain and the body.

## Data Availability

Not applicable.

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
