# Peer review of "Varied Composition and Underlying Mechanisms of Gut Microbiome in Neuroinflammation"

_microorganisms, 2022, doi:10.3390/microorganisms10040705_

Round 1
Reviewer 1 Report
The manuscript entitled „Varied Composition and Underlying Mechanisms of Gut Microbiome in Neuroinflammation” presents interesting issue, but some issues should be corrected.
General:
Authors should use a proper scientific vocabulary and present science-based facts and opinions, as sentences such as “Ancient myths associate future personality of a new born child with the person who administered them with a chewed and softened food” may be rather adequate for a column of newspaper, but not for a scientific manuscript.
There are really important problems associated with the prepared manuscript, including:
- The serious flaw of the presented manuscript is associated with the fact, that it presents a highly subjective review, not a systematic review. While the systematic review has a key role for broadening knowledge, the other reviews don’t have such role.
- Taking into account, that the Materials and methods section is not presented (it should be added), without any specific information, it is hard to understand which studies were included into review and why. Authors did not present any key words, which were used during literature search, inclusion and exclusion criteria of references, information about the procedure of literature search conducted by them, number of chosen references, as well as information if some of them were excluded from the review and on the basis of which criteria. As a number of recent publications that are related to the issue were not included, it is a serious problem.
- Authors do not present the current and comprehensive knowledge associated with the issue. It is associated with the fact that they did not include some important issues, while other were included even if they are not so crucial (e.g. sub-chapter “Microbiome Project in South Korea”)
- The manuscripts presents a number of basic and even trivial information, that are well known not only for the readers of journal but also for everybody (e.g. “definition of inflammation).
Authors should prepare the manuscript not only to be interesting for Korean readers, but to be interesting for international readers. If Authors prepare their manuscript only for their national readers, they should publish it in some national journal. So, Authors should present here international data from various countries, not only the Korean ones. If Authors decide to present information specific for South Korea (sub-chapter “Microbiome Project in South Korea”), they should present adequate information also for the other countries, while presenting international perspective and similar projects also from the other countries.
Author Response
|
# |
Comments from reviewers |
Response from authors |
|
|
Reviewer #1 |
Authors should use a proper scientific vocabulary and present science-based facts and opinions, as sentences such as “Ancient myths associate future personality of a new born child with the person who administered them with a chewed and softened food” may be rather adequate for a column of newspaper, but not for a scientific manuscript. |
Thank you for your comment. The mentioned statement in the abstract has been replaced by a more scientifically relevant statement given as under, also highlighted in the abstract.
“Human gut microbiome has been implicated in a host of bodily functions and their regulation, including brain development and cognition”. |
|
|
· The serious flaw of the presented manuscript is associated with the fact, that it presents a highly subjective review, not a systematic review. While the systematic review has a key role for broadening knowledge, the other reviews don’t have such role. |
Thank you for highlighting this. We would like to emphasize again that this has been intentionally drafted as a subjective review and systematic review of the subject was not within the scope of this project. |
|
|
|
· Taking into account, that the Materials and methods section is not presented (it should be added), without any specific information, it is hard to understand which studies were included into review and why. Authors did not present any key words, which were used during literature search, inclusion and exclusion criteria of references, information about the procedure of literature search conducted by them, number of chosen references, as well as information if some of them were excluded from the review and on the basis of which criteria. As a number of recent publications that are related to the issue were not included, it is a serious problem. |
For the reason stated above that it is not a systematic review, a PRISMA figure, inclusion exclusion criteria and materials and methods section is not formulated nor is presented.
However, in the light of your advice, latest publication on different aspects of the subject under review are included in the draft. The mentioned changes are made in track changes and highlighted in color. This has added value to the manuscript, and we are grateful for this suggestion. |
|
|
|
· Authors do not present the current and comprehensive knowledge associated with the issue. It is associated with the fact that they did not include some important issues, while other were included even if they are not so crucial (e.g. sub-chapter “Microbiome Project in South Korea”) |
Thank you for highlighting the fact that we added the chapter on Korean research. The reason is that this submission is a part of this special issue of the journal focused on Korean research on microbiome.
However in the light of your advice, we have added latest references to the subject that might have been missed previously, including references numbers 14, 19, 345, and 346. These references are inserted in color to be noted. |
|
|
|
· The manuscripts presents a number of basic and even trivial information, that are well known not only for the readers of journal but also for everybody (e.g. “definition of inflammation). |
Thank you for the comment. The purpose of presenting definitions is to introduce the reader to the operational criteria under which the subject is being discussed. There can be various definitions, so an effort has been made to present the information and operational definitions. |
|
|
|
Authors should prepare the manuscript not only to be interesting for Korean readers, but to be interesting for international readers. If Authors prepare their manuscript only for their national readers, they should publish it in some national journal. So, Authors should present here international data from various countries, not only the Korean ones. If Authors decide to present information specific for South Korea (sub-chapter “Microbiome Project in South Korea”), they should present adequate information also for the other countries, while presenting international perspective and similar projects also from the other countries. |
Once again thank you for your comment. The manuscript in general is intended for an international audience and all the literature related to the topic has been included which will attract the broad readership of the esteemed journal globally. However, the chapter on the Korean research on the subject has been added because of the special issue of the journal focuses on the Korean research on this subject. A bibliometric review of the subject, which will measure research on this subject from every country individually may be an interesting idea and we are grateful to you for highlighting this to us. |
|
Reviewer 2 Report
This is a well written, interesting review article that will contribute to the literature.
Author Response
|
Reviewer #2 |
This a comprehensive and well-organized review around the role of gut microbiota and neuroinflammation. It is an informative work and inclusive, and therefore I recommend its publication after minor changes. As a reader in the area, I do find it really interesting review and I would recommend reading it. However, my main concern is around the discussion of the ritual of ghutti or tahnik which may be understood as beneficial from the context despite the author mentioned that “this ritual is largely give up for hygiene and appropriateness…..”. I think adding some phrases such as “there is no evidence so far that there is any correlation between the gut microbiome composition and the oral microbiome of the person who is doing the ritual” would clarify the narrative from the scientific facts. On the contrary, the oral microbiome seems to only negatively associated with gut microbiome and periodontal diseases seem to cause gut microbial dysbiosis. I am also highly concerned about the first part of figure 3 but only for the pre-microbiome section which is again proposes directly that ghutti or vaginal swab are needed with no evidence whatsoever that neither of these is really needed, and the risk of both practices are higher than any benefits especially if practiced outside a hospital. The lower part starting from “minimize risk….” Is a reasonable suggestion. |
Many thanks for your valuable feedback. In light of your advice, the mentioned statement has been changed to a more scientifically accurate expression as follows.
We agree that there is no empirical evidence so far that there is any correlation between the gut microbiome composition and the oral microbiome of the person who is doing the ritual and the practice has largely been given up probably for concerns about hygiene and appropriateness for the baby.
In addition to the above, the figure stating the possible points of intervention has also been updated as follows.
Establish a scientifically safe way to seed healthy flora in a newborn’s GI tract.
We are grateful for these suggestions as they have enhanced the precision of scientific expression of the draft.
|
|
Another point to consider is that figure 2 does not feature the diet which is also influential in shaping the gut microbiome; it cold be added as an item under primary colonisation. |
Thank you for your suggestion. The suggested item, diet, has been added under the primary colonization in figure 2. |
|
|
I also suggest a language check as there are minor linguistic changes required, and I will name just some visible ones: · Line 25, allotransplantation is repeated. · Line 36-37, mounting impact and high impact are just synonyms, and so one of them is enough. · Line 100, delete Diseases · Line 137, “is considered” rather then “considers” · Line 177, “Same Way” should changed to “same way”
|
· Thank you again, modified as suggested · · Line 25, allotransplantation is removed.
Line 36-37, corrected
Line 100, diseases deleted Line 137, “considers” changed to “is considered” Corrected as advised |
Reviewer 3 Report
Dear Editor,
This a comprehensive and well-organized review around the role of gut microbiota and neuroinflammation. It is an informative work and inclusive, and therefore I recommend its publication after minor changes. As a reader in the area, I do find it really interesting review and I would recommend reading it. However, my main concern is around the discussion of the ritual of ghutti or tahnik which may be understood as beneficial from the context despite the author mentioned that “this ritual is largely give up for hygiene and appropriateness…..”. I think adding some phrases such as “there is no evidence so far that there is any correlation between the gut microbiome composition and the oral microbiome of the person who is doing the ritual” would clarify the narrative from the scientific facts. On the contrary, the oral microbiome seems to only negatively associated with gut microbiome and periodontal diseases seem to cause gut microbial dysbiosis. I am also highly concerned about the first part of figure 3 but only for the pre-microbiome section which is again proposes directly that ghutti or vaginal swab are needed with no evidence whatsoever that neither of these is really needed, and the risk of both practices are higher than any benefits especially if practiced outside a hospital. The lower part starting from “minimize risk….” Is a reasonable suggestion.
Another point to consider is that figure 2 does not feature the diet which is also influential in shaping the gut microbiome; it cold be added as an item under primary colonisation.
I also suggest a language check as there are minor linguistic changes required, and I will name just some visible ones:
- Line 25, allotransplantation is repeated.
- Line 36-37, mounting impact and high impact are just synonyms, and so one of them is enough.
- Line 100, delete Diseases
- Line 137, “is considered” rather then “considers”
- Line 177, “Same Way” should changed to “same way”
This a comprehensive and well-organized review around the role of gut microbiota and neuroinflammation. It is an informative work and inclusive, and therefore I recommend its publication after minor changes. As a reader in the area, I do find it really interesting review and I would recommend reading it. However, my main concern is around the discussion of the ritual of ghutti or tahnik which may be understood as beneficial from the context despite the author mentioned that “this ritual is largely give up for hygiene and appropriateness…..”. I think adding some phrases such as “there is no evidence so far that there is any correlation between the gut microbiome composition and the oral microbiome of the person who is doing the ritual” would clarify the narrative from the scientific facts. On the contrary, the oral microbiome seems to only negatively associated with gut microbiome and periodontal diseases seem to cause gut microbial dysbiosis. I am also highly concerned about the first part of figure 3 but only for the pre-microbiome section which is again proposes directly that ghutti or vaginal swab are needed with no evidence whatsoever that neither of these is really needed, and the risk of both practices are higher than any benefits especially if practiced outside a hospital. The lower part starting from “minimize risk….” Is a reasonable suggestion.
Another point to consider is that figure 2 does not feature the diet which is also influential in shaping the gut microbiome; it cold be added as an item under primary colonisation.
I also suggest a language check as there are minor linguistic changes required, and I will name just some visible ones:
- Line 25, allotransplantation is repeated.
- Line 36-37, mounting impact and high impact are just synonyms, and so one of them is enough.
- Line 100, delete Diseases
- Line 137, “is considered” rather then “considers”
- Line 177, “Same Way” should changed to “same way”
Author Response

(The authors gave the same response as above.)

Round 2
Reviewer 1 Report
The manuscript entitled „Varied Composition and Underlying Mechanisms of Gut Microbiome in Neuroinflammation” presents interesting issue, but some issues should still be corrected.
There are really important problems associated with the prepared manuscript, including:
- The serious flaw of the presented manuscript is associated with the fact, that it presents a highly subjective review, not a systematic review. While the systematic review has a key role for broadening knowledge, the other reviews don’t have such role.
- Taking into account, that the Materials and methods section is not presented (it should be added), without any specific information, it is hard to understand which studies were included into review and why. Authors did not present any key words, which were used during literature search, inclusion and exclusion criteria of references, information about the procedure of literature search conducted by them, number of chosen references, as well as information if some of them were excluded from the review and on the basis of which criteria. As a number of recent publications that are related to the issue were not included, it is a serious problem.